# Breadth of CD8 T-cell mediated inhibition of replication of diverse HIV-1 transmitted-founder isolates correlates with the breadth of recognition within a comprehensive HIV-1 Gag, Nef, Env and Pol potential T-cell epitope (PTE) peptide set

**Peter Hayes**[1]*, **Natalia Fernandez**[1], **Christina Ochsenbauer**[2], **Jama Dalel**[1], **Jonathan Hare**[1], **Deborah King**[1], **Lucas Black**[1], **Claire Streatfield**[1], **Vanaja Kakarla**[1], **Gladys Macharia**[1], **Julia Makinde**[1], **Matt Price**[3,4], **Eric Hunter**[5], **The IAVI protocol C investigators**[¶], **Jill Gilmour**[1]

**1** IAVI Human Immunology Laboratory, Imperial College, London, United Kingdom, **2** University of Alabama, Birmingham, Alabama, United States of America, **3** IAVI, New York, New York, United States of America, **4** Department of Epidemiology and Biostatistics, University of California at San Francisco, San Francisco, California, United States of America, **5** Emory Vaccine Center, Atlanta, Georgia, United States of America

¶ Membership of the IAVI protocol C investigators is listed in the Acknowledgments.
* p.hayes@imperial.ac.uk

## Abstract

Full characterisation of functional HIV-1-specific T-cell responses, including identification of recognised epitopes linked with functional antiviral responses, would aid development of effective vaccines but is hampered by HIV-1 sequence diversity. Typical approaches to identify T-cell epitopes utilising extensive peptide sets require subjects' cell numbers that exceed feasible sample volumes. To address this, CD8 T-cells were polyclonally expanded from PBMC from 13 anti-retroviral naïve subjects living with HIV using CD3/CD4 bi-specific antibody. Assessment of recognition of individual peptides within a set of 1408 HIV-1 Gag, Nef, Pol and Env potential T-cell epitope peptides was achieved by sequential IFNγ ELISpot assays using peptides pooled in 3-D matrices followed by confirmation with single peptides. A *Renilla reniformis* luciferase viral inhibition assay assessed CD8 T-cell-mediated inhibition of replication of a cross-clade panel of 10 HIV-1 isolates, including 9 transmitted-founder isolates. Polyclonal expansion from one frozen PBMC vial provided sufficient CD8 T-cells for both ELISpot steps in 12 of 13 subjects. A median of 33 peptides in 16 epitope regions were recognised including peptides located in previously characterised HIV-1 epitope-rich regions. There was no significant difference between ELISpot magnitudes for in vitro expanded CD8 T-cells and CD8 T-cells directly isolated from PBMCs. CD8 T-cells from all subjects inhibited a median of 7 HIV-1 isolates (range 4 to 10). The breadth of CD8 T-cell mediated HIV-1 inhibition was significantly positively correlated with CD8 T-cell breadth of peptide recognition. Polyclonal CD8 T-cell expansion allowed identification of HIV-1 isolates inhibited and peptides recognised within a large peptide set spanning the major HIV-1

**Data Availability Statement:** All relevant data are within the paper and its Supporting information files.

**Funding:** This work was made possible by IAVI, which is supported by funding from many donors, including the Bill and Melinda Gates Foundation, the Ministry of Foreign Affairs of Denmark, Irish Aid, the Ministry of Finance of Japan in partnership with The World Bank, the Ministry of Foreign Affairs of the Netherlands, the Norwegian Agency for Development Cooperation, the United Kingdom Department for International Development and the US Agency for International Development. This work is made possible by the generous support of the American people through USAID. The contents are the responsibility of the authors and do not necessarily reflect the views of USAID or the United States Government. A full list of IAVI donors is available at: http://www.iavi.org. The funding donors had no role in the study design, data collection, and analysis, decision to publish, or preparation of the manuscript.

**Competing interests:** The authors have declared that no competing interests exist.

proteins. This approach overcomes limitations associated with obtaining sufficient cell numbers to fully characterise HIV-1-specific CD8 T-cell responses by different functional readouts within the context of extreme HIV-1 diversity. Such an approach will have useful applications in clinical development for HIV-1 and other diseases.

# Introduction

Effective protection against human immunodeficiency virus-1 (HIV-1) infection is likely to require both humoral and cellular-mediated immune responses [1–4] with cytotoxic CD8 T-cells being a key component of the cellular immune response to HIV-1 infection [4–6]. The ability to assess T-cell mediated inhibition of replication of diverse HIV-1 isolates and link this with recognition of individual viral epitopes, would aid the understanding of potential correlates of immune control and identification of broadly effective T-cell targets, thereby providing tools for rational T-cell immunogen design and effective vaccine candidates.

Simultaneous and in-depth assessments of multiple CD8 T-cell functions and in particular the detailed mapping of HIV-1 epitopes recognised, is hampered by the immense HIV-1 sequence diversity [7]. Detailed mapping of individual epitopes recognised by T-cells across the HIV-1 proteome would require extensive peptide sets, typically tested in interferon gamma (IFNγ) enzyme-linked immunospot (ELISpot) assay utilising subjects' peripheral blood mononuclear cells (PBMC). Such peptide sets have been designed based on a consensus of the most common amino acid present at each site across multiple HIV-1 protein sequences [8, 9]. However, this consensus approach may exclude many epitopes recognised by T-cells [10]. The use of autologous peptides based on the HIV-1 sequence of each subject would identify more epitopes per subject [6, 10], but would not be practicable in studies of multiple subjects, requiring both HIV-1 sequence information and a unique peptide set matched to each subject. One approach designed to reduce the number of peptides tested, whilst still addressing HIV-1 sequence diversity and not requiring prior knowledge of subjects' HIV-1 sequences is the use of potential T cell epitopes (PTE). Such sets of 15 amino acid (15mer) peptides have been designed to include the most frequent naturally occurring 9 amino acid epitopes present within Gag, Nef, Env and Pol proteins within the sequences of HIV-1 circulating worldwide [11, 12] and are available from the NIH HIV reagent program. However, even this set consists of 1408 peptides, each to be assessed for T-cell recognition. Rather than testing each individual peptide, arrangement of peptides into pool matrices have been used to reduce the number of tests and PBMC required to identify CD8 T-cell specificities [13–15]. Peptides are arranged such that each is contained in more than one peptide pool or dimension, typically in two- or three-dimensional matrix pools. A first IFNγ ELISpot assay determines matrix pool responses which often only narrows down the possible peptides recognised, especially where a subject has multiple epitope responses within each HIV-1 protein. A second assay is then conducted to confirm individual peptides recognised. However, combining the PTE and matrices approaches to identify individual peptides recognised by T-cells across the entire HIV-1 Gag, Nef, Env and Pol proteome, would still require blood sampling volumes from subjects that would far exceed what would be typically acceptable for most clinical studies and study participants. Assessment of additional functional T-cell responses in flow cytometric and HIV-1 inhibition assays would further add to cell requirements.

One approach to addressing the issue of sample constraints involves polyclonal expansion of T-cells in vitro. Both CD4 and CD8 T-cells can be polyclonally expanded by culture of

PBMC with CD3/CD8 or CD3/CD4 bi-specific antibodies respectively in the presence of interleukin-2 (IL-2), yielding large numbers of relatively pure (>90%) CD4 or CD8 T-cells [16]. Polyclonal T-cell expansion has been employed in HIV-1 viral inhibition assays (VIA) that assess the ability of CD8 T-cells to inhibit HIV-1 replication in infected autologous CD4 T-cells and has been applied to clinical trials of HIV-1 vaccine candidates and studies of natural infection [17–19]. An optimised VIA assay [20] uses replication competent HIV-1 infectious molecular clones (IMC) engineered with a novel *Renilla reniformis* luciferase gene reporter technology (LucR) [21–23]. A potential criticism of the polyclonal expansion approach is that the functional profiles of polyclonally expanded T-cells may not be directly comparable with those of *ex vivo* uncultured T-cells. This was found not to be the case for CD8 T-cell mediated inhibition of HIV-1 replication in the VIA, where inhibition of HIV-1 replication was equivalent when both polyclonally expanded CD8 T-cells and CD8 T-cells isolated directly from PBMC were assessed [19]. A recent report from our group also found similar IFNγ ELISpot responses to HIV-1 Gag peptides in directly isolated and polyclonally expanded CD8 T-cells [24].

This report describes an approach to generate sufficient T-cells by polyclonal expansion from 1 cryopreserved PBMC vial (~1 to $2 \times 10^7$ PBMC frozen) to allow assessment of CD8 T-cell mediated inhibition of HIV-1 replication in autologous CD4 T-cells and identification of individual peptides recognised by CD8 T-cells using a set of 1408 Gag, Nef, Env and Pol PTE peptides. Peptide-specific responses for polyclonally expanded and unexpanded CD8 T-cells were similar, as were expanded and then cryopreserved CD8 T-cells. For CD8 T-cell mediated inhibition studies, a multi-clade HIV-1 panel was employed that included 9 IMC derived from subjects with recently acquired HIV-1 (transmitted-founder isolates) [25]. CD8 T-cell breadth of peptide recognition and inhibition of HIV-1 replication were positively correlated. This approach allows for the simultaneous capture of multiple CD8 T-cell functions from a limited number of subject sample vials, yielding valuable information to support clinical investigations and product development.

## Methods

### Subjects and PBMC isolation

Subjects were drawn from a large prospective cohort of 613 volunteers living with HIV to evaluate clinical, laboratory, immunologic and viral markers of disease progression (IAVI protocol C) [26]. From this cohort, 13 volunteers from clinics in Uganda, Rwanda, Zambia and South Africa were selected based on being i) apparently healthy in clinical latency, ii) anti-retroviral (ARV) therapy naïve and iii) willing to donate up to 100mL of blood at an additional visit which ranged at between 1203 and 5319 days post estimated date of HIV-1 acquisition (EDA) (Table 1). Blood samples from HIV-1 uninfected South African blood transfusion volunteers were also obtained through Contract Laboratory Services Ltd., Johannesburg, South Africa. Peripheral blood mononuclear cells (PBMC) were isolated by density centrifugation, frozen at 10 million viable PBMC per vial in 10% V/V dimethyl sulphoxide (DMSO) in heat-inactivated foetal calf serum (HIFCS) (all Merck Life Science Ltd., UK) and stored in vapour phase liquid nitrogen.

### PTE peptides and mapping strategy

Four pools of HIV-1 Gag, Nef, Env and Pol PTE 15mer peptides and 1408 individual peptides were obtained through the NIH HIV Reagent Program, Division of AIDS, NIAID, NIH, Germantown, USA, contributed by DAIDS/NIAID. Peptides were dissolved in DMSO such that each peptide once pooled was at a concentration of 222μg/mL and stored at -80°C. Individual

**Table 1. Information for subjects living with HIV.**

| Subject | Gender (F / M) & age (years) | Country | HIV-1 clade | Plasma viral load/mL at sample time point | CD4/µL blood at sample time point | Days post EDA |
|---|---|---|---|---|---|---|
| 1 | F 36 | Rwanda | A | 39 | 581 | 2691 |
| 2 | M 22 | Rwanda | A | 29463 | 547 | 3033 |
| 3 | F 32 | Uganda | A | 1353 | 412 | 1736 |
| 4 | M 44 | Rwanda | A | 1620 | 519 | 1602 |
| 5 | F 40 | Uganda | A | 2350 | 374 | 5319 |
| 6 | F 35 | Zambia | C | 141815 | 328 | 2842 |
| 7 | F 36 | Zambia | C | 18513 | 665 | 1698 |
| 8 | F 24 | South Africa | C | 305 | 604 | 1412 |
| 9 | F 21 | South Africa | C | 205 | 566 | 2358 |
| 10 | M 23 | South Africa | C | 17718 | 389 | 2402 |
| 11 | F 30 | Zambia | C | 1823 | 708 | 1203 |
| 12 | F 37 | Uganda | D | 165 | 674 | 2530 |
| 13 | M 31 | Uganda | D | 2977 | 356 | 2208 |
| | | | Mean | 16796 | 517 | 2387 |
| | | | Median | 1823 | 547 | 2358 |
| | | | Min | 39 | 328 | 1203 |
| | | | Max | 141815 | 708 | 5319 |

peptides were combined into a total of 174 pools of 18 to 26 peptides per pool such that each individual peptide was present in three different peptide pools: a 3-dimensional (3-D) peptide matrix. Matrix pools remained separated into matrices for each of the four HIV-1 proteins (Table 2).

Peptides were diluted in RPMI medium with 10% V/V heat-inactivated calf serum (HIFCS) (R10) and used at a final ELISpot assay concentration of 1µg/mL per peptide and <0.5% V/V DMSO. Duplicate peptide pools and quadruplicate DMSO/R10 only wells as negative controls (mock) were stored frozen in 96 well deep well plates in the arrangement required for IFNγ ELISpot assay. Phytohemagglutinin (PHA) (Merck Life Science Ltd., UK) as a positive control was prepared and added at the point of ELISpot assay set up at 10µg/mL final assay concentration.

## Expansion of CD8 T-cells and IFNγ ELISpot assay

CD8 T-cells were expanded from PBMC and IFNγ ELISpot assays performed using a method similar to that described elsewhere [24]. Typically, 6 to 12 million thawed viable PBMC were

**Table 2. PTE peptide sets and peptide pool details.** Sets of individual 15mer peptides and pools of peptides for HIV-1 Gag, Nef, Env and Pol proteins were supplied by the NIH HIV Reagent program, USA.

| HIV-1 protein | Individual peptide set catalogue number | Pooled peptide set catalogue number | PTE peptides per set | Number of matrix pools (total / per dimension) | PTE peptides per matrix pool |
|---|---|---|---|---|---|
| Gag | 11554 | 12437 | 320 | 39 / 13 | 22–27 |
| Nef | 11553 | 12822 | 127 | 21 / 7 | 16–20 |
| Env | 11551 | 12698 | 480 | 57 / 19 | 23–27 |
| Pol | 11552 | 12961 | 480 | 57 / 19 | 23–27 |
| Total | | | 1408 | 174 | |

resuspended in 10mL R10 with 50 units IL-2/mL (R10/50) and 0.5µg/mL CD3/CD4 bispecific antibody (kindly provided by Professor Johnson Wong, Harvard Medical School, USA). Cultures were doubled in volume with R10/50 at days 2 and 4, transferring cultures to additional flasks.

On day 7, approximately 2/3 of or up to 100 million cells were removed, washed 3 times with 20mL PBS and resuspended in RPMI medium with 20% HIFCS (R20) without IL-2 at half the removed culture volume. Cells were rested for 22 to 26 hrs and IFNγ ELISpot assay conducted on day 8 as described previously [27, 28]. In brief, 100µL mock (DMSO/R10), 3-D matrix HIV-1 PTE peptide pools and PHA were added to IFNγ ELISpot plates (Mabtech AB, Sweden) followed by 50µL cells (1 to $2x10^5$ cells/well). Following 16 to 24 hours incubation (day 9), spots were developed by sequential incubation with biotinylated anti-human IFNγ antibody (Mabtech AB, Sweden), Streptavidin/biotin/peroxidase complex (Life Technologies Ltd., UK) and finally AEC/buffer peroxidase substrate (Merck Life Science Ltd., UK), with wells washed with PBS/0.5% v/v tween-20 (Merck Life Science Ltd., UK) between steps. Spots were enumerated using an ELISpot reader (AID Autoimmun Diagnostika, Germany). During days 10 and 11, individual peptides to test from the complete 1408 peptide set were identified by deconvolution of the responding peptide matrix pools and diluted in R10 in the required 96 well plate format. Peptides were added to wells of IFNγ ELISpot plates on day 11.

Also, on days 7 and 9, the remaining 1/3 of culture was doubled in volume with R10/50. On day 10, cultures were recovered, washed three times with PBS and rested as above. On day 11, cells were recovered, counted and added to ELISpot wells as before to assess CD8 T-cell responses to selected individual PTE peptides. Any remaining cells were resuspended in chilled (+2 to +8˚C) 10% DMSO/90% HIFCS and 1 mL aliquots with up to $30x10^6$ cells per vial were cryopreserved in liquid nitrogen. Frozen cells were thawed and washed in R10 medium, rested overnight in R20 medium before testing in IFNγ ELISpot assay.

For an IFNγ ELISpot assay to be considered valid, the mean of mock wells must be ≤50 and for PHA >500 spot forming units (SFU) per million cells. A response to a peptide or peptide pool stimulus was considered positive if the mean of replicate wells was ≥4 times the mean mock response and the mean mock-subtracted response was ≥38 SFU per million cells [24, 27, 28].

A list of well-defined HIV-1 CD8 T-cell epitopes was acquired from the LANL HIV immunology database: http://www.hiv.lanl.gov/content/immunology (HIV Sequence Compendium 2018). This list, often referred to as the "A-list" consists of 281 epitopes across HIV-1 proteins and includes the sequence of each epitope and its location on the HIV-1 HXB2 protein sequence [29]. All sequences for peptide responses identified in the present study and the A-list peptides were collated and aligned against the HXB2 protein sequence. Alignments were performed using the protein sequence alignment tool from: https://www.ncbi.nlm.nih.gov/ (National Center for Biotechnology Information (NCBI)).

## Isolation of CD8 T-cells from PBMC and IFNγ ELISpot assay

To compare ELISpot responses of expanded CD8 T-cells and CD8 T-cells isolated directly from PBMC without expansion, 6 of the 13 subjects living with HIV were selected (Table 1 subjects 1, 6, 8, 9, 10 & 13) based on additional frozen PBMC vial availability and positive responses to Nef peptide pools. CD8 T-cells were expanded from PBMC for 10 days and washed as before. Based on prior known PBMC recoveries per vial, 4 to 6 vials from each of these 6 subjects were then thawed and both expanded CD8 T-cells and PBMC were rested for 23 hrs in R20. So that all CD8 T-cells underwent similar conditions just prior to ELISpot assay, CD8 T-cells were isolated from all cell cultures by negative selection using a CD8 T-cell

isolation kit (Miltenyi Biotech Ltd., UK) following manufacturer's instructions. Cells were tested in IFNγ ELISpot assay as before, with 200,000 cells per well with one of the following stimuli; mock, PHA, a pool of 138 15mer peptides overlapping by 11 amino acids matched to the *Cytomegalovirus* (CMV) pp65 protein (Genscript Biotech, Netherlands) or the 7 Nef peptide pools of the 1st dimension of the Nef matrix. Peptides were again used at a final assay concentration of 1µg/mL per peptide. Nef was selected as the PTE set with the fewest peptides, a full Nef and/or other HIV-1 protein matrix ELISpot assay could not be conducted due to the limits on frozen PBMC vial availability and CD8 T-cell recovery from PBMC.

## Flow cytometric analyses of expanded and PBMC CD8 T-cells

In a separate study [30] of HIV-1 pathogenesis currently in progress within the IAVI protocol C cohort, IFNγ production by CD8 T-cells in response to CMVpp65 or HIV-1 Gag, Nef, Env or Pol PTE peptide pool stimuli was assessed by intra-cellular cytokine flow cytometry. Both thawed and overnight rested PBMC and 7-day expanded, washed and rested CD8 T-cells from the same subject samples (n = 28) were assessed for % CD3+CD8+ T-cells expressing intra-cellular IFNγ.

Thawed and overnight rested PMBCs and expanded and rested CD8 T-cells were washed in R10 media and 0.75–1.0 $1x10^6$ cells were stimulated for 6 hours with HIV-1 Gag, Nef, Env, Pol PTE or CMVpp65 peptide set pools along with mock and positive control Staphylococcal enterotoxin B, (SEB, Merck Life Science Ltd., UK) in the presence of 12.5µg/mL Brefeldin A (Merck Life Science Ltd., UK) and 0.002mM monensin (BioLegend Ltd., UK). Cells were washed in PBS, stained with LIVE/DEAD Fixable Near-IR dead cell stain (Life Technologies Ltd., UK) according to manufacturer's instructions, washed with 2% V/V HIFCS in PBS and incubated with anti-CD3 SK7, anti-CD8 RPA-T8, anti-CD4 SK3 and anti-CD19 HIB19 (Becton Dickinson, San Jose, USA) monoclonal antibodies at 4˚C in the dark for 30min. Cells were then permeabilised with BD Cytofix/Cytoperm kit (Beckton Dickinson, USA) according to the manufacturer's instructions and incubated with anti-IFNγ B27 antibody (Becton Dickinson, USA) at 4˚C in the dark for 30 minutes and washed. S1 Table provides details of antibodies and dilutions used. Cells were analysed using a FACSymphony flow cytometer and data were analyzed and presented using FlowJo software version 10.6 (Becton Dickinson, USA).

## HIV-1 proviral plasmid construction: IRES constructs

Generation of HIV-1 IMC [21, 31] and the optimisation of replicating *Renilla* luciferase reporter HIV-1 utilising a modified encephalomyocarditis virus (EMCV) internal ribosome entry site (IRES) element, collectively referred to as IMC-LucR.6ATRi [22, 23] has been described previously. All plasmids were generously provided by Dr Christina Ochsenbauer, University of Alabama, Birmingham, USA (Table 3). All but virus number 4 (NL-LucR) were derived from subjects living with recently acquired HIV-1 (transmitted-founder isolates).

## Phylogenetic analysis of HIV-1 isolates

Single genome near full length nucleotide sequences for each viral gene (Gag, Pol, Vif, Vpr, Vpu, Tat, Rev, Env and Nef) were translated to amino acid sequences using the Los Alamos National Laboratory genecutter tool (https://www.hiv.lanl.gov/content/sequence/GENE_CUTTER/cutter.html). The translated proteomes were concatenated and aligned on Geneious Prime version 2020.0.3 followed by phylogenetic analysis in MEGA version X [32, 33]. The initial phylogenetic tree for the heuristic search was obtained by applying the Maximum Parsimony method after which the evolutionary history was inferred by using the Maximum Likelihood method and General Reverse Transcriptase model that allowed for some sites to be

**Table 3. Details of LucR.6ATRi IMC reporter viruses tested in the viral inhibition assay.**

| Virus | HIV-1 LucR.6ATRi IMC | Clade | Country of origin | Genbank accession |
|---|---|---|---|---|
| 1 | A1.R6185M-21 | A1 | Rwanda | KC018956 |
| 2 | A1.R3469F | A1 | Rwanda | KC018937 |
| 3 | A1/D.191947 | A1/D | Uganda | KC018999 |
| 4 | NL | B | USA | AF324493 |
| 5 | B.CH077 | B | USA | GQ925949 |
| 6 | C.Z3618M | C | Zambia | KR820366 |
| 7 | C.Z3678M | C | Zambia | KR820393 |
| 8 | C.ZM247F_V2 | C | Zambia | EU166832 |
| 9 | C.Z1123M | C | Zambia | KP715741 |
| 10 | D.191882 | D | Uganda | KC018572 |

evolutionarily invariable [34]. 100 permutations were performed and a tree with the highest log likelihood selected. The tree was drawn to scale, with branch lengths measured in the number of substitutions per site. Pairwise distances between pairs of sequences were derived using the General Reverse Transcriptase model involving 222 amino acid sequences and a total of 3429 positions.

## Cell lines and generation of LucR IMC

HEK 293T/17 cells were obtained directly from the American Type Culture Collection (ATCC, Manassas, VA, USA) (CRL-11268). TZM-bl dual-reporter cell line was obtained directly from the NIH HIV Reagent program, Division of AIDS, NIAID, NIH: TZM-bl Cells, ARP-8129, contributed by Dr. John C. Kappes and Dr. Xiaoyun Wu. Both cell lines were maintained in Dulbecco's modified Eagle's medium (DMEM) with 10% V/V HIFCS, 10 mM hepes, 100 U/mL penicillin and streptomycin (all Merck Life Science Ltd., UK) at 37°C and 5% $CO_2$.

Virus stocks of HIV-1-infectious molecular clones were generated by proviral DNA transfection of 293T/17 cells as previously described [21, 22, 31, 35]. Briefly, $5 \times 10^6$ 293T/17 cells were seeded in a T75 culture flask one day prior to transfection. 12μg of DNA in DMEM was transfected using FuGENE 6 (Promega Ltd., UK), according to manufacturer's instructions. After 6 hr, transfection medium was replaced with fresh medium. Viral supernatants were harvested from 293T/17 cells at 48 hrs, supernatants clarified at 250g/10 min, 0.45μm filtered and frozen at -80°C.

The fifty-percent Tissue-Culture Infectious Dose ($TCID_{50}$) per mL of virus stocks were determined as previously described [35]. Serial dilutions of viruses were added to flat bottom 96 well plates in quadruplicate followed by 100,000 TZM-bl cells in the presence of 10μg DEAE-dextran (Merck Life Science Ltd., UK) per mL. Plates were incubated at 37°C and 5% $CO_2$ for 48 hr. 100μL culture medium was carefully removed from each well and replaced with luciferase reporter gene assay system reagent (BriteLite Plus Assay System, PerkinElmer Ltd., UK). After 3 min incubation, luminescence was measured using a Tecan Infinite® reader (Tecan Ltd., Switzerland).

## LucR IMC viral inhibition assay

CD8 T-cells were polyclonally expanded from PBMC as above for 7 days. CD4 T-cells were expanded in a similar manner with 0.5 μg/mL CD3/CD8 bispecific antibody and IL-2 (Professor Johnson Wong, Harvard Medical School, USA). Cultures volumes were doubled at days 3 and 6 with R10/50.

On day 7, $1x10^6$ viable CD4 T-cells were centrifuged in 15mL tubes at 250g/10 min and supernatants decanted with most of the residual volume aspirated. Cells were resuspended in the remaining residual volume and one of ten HIV-1 LucR.6ATRi IMC viruses added at a multiplicity of infection (MOI) of 0.1 (Table 3). Cells and virus were spinoculated at 1800g for 2 hours at ambient temperature and cells resuspended at $1x10^6$ cells/mL in R10/50. 100 μL CD4 T-cells (100,000 cells) were placed in quadruplicate in 96-well flat bottom white plates. On the same day, expanded CD8 T-cell cells were resuspended at $1x10^6$ cells/mL in fresh R10/50 and cultured in 24-well culture plates. Both the CD8 and infected CD4 T-cells were incubated in parallel for 3 days, when CD8 T-cells were recovered from wells, centrifuged, supernatants decanted and cells resuspended in an equal volume of R10/50. 100μL CD8 T-cells or R10/50 media were added to the infected CD4 T-cells in duplicate and incubated for an additional 5 days. On day 8 post-CD4 T-cell infection, 100 μL of supernatant were carefully removed from wells and 100 μL Renilla-Glo™ Luciferase Assay System (Promega Ltd., UK) added. Plates were incubated for 3 min at room temperature, followed by measurement of luminescence activity using a Tecan® Infinite reader. Luciferase activity was determined as a measure of viral infection and replication measured in relative light units (RLU). CD8 T-cell-mediated inhibition was expressed as reduction in $\log_{10}$ RLU of CD4/CD8 T-cell co-cultures, compared with cultures of infected CD4 T-cells alone. A value of $\geq 0.8 \log_{10}$ inhibition was considered positive [20].

## Assessment of HIV-1 replication in expanded CD8 T-cells from subjects living with HIV

In the separate study of HIV-1 pathogenesis currently in progress within the IAVI protocol C cohort [30], culture supernatants were sampled from day 10 expanded CD8 T-cells of 71 ARV-naïve subjects living with HIV (median pVL 13600/mL, range <39 to $1.46x10^6$, IQR 1995 to 92500). In order to assess possible productive replication of subjects' own HIV-1 in expanded CD8 T-cells, supernatant p24 gag protein contents were determined by commercial ELISA following the supplier provided protocol (Perkin Elmer Ltd., UK, p24 ELISA NEK050). As a comparison with cultures where productive HIV-1 replication is expected, day 13 culture supernatants were included from 7 day expanded CD4 T-cells from a HIV-1 uninfected subject that were infected in vitro with one of three HIV-1 isolates at a MOI of 0.01. The isolates were not HIV-1 LucR molecular constructs but were analogous to the NL, CH077 and 247F_V2 constructs described in Table 3.

## Statistics

Data were analysed using GraphPad Prism version 8.4.3 for Windows, GraphPad Software, San Diego, USA.

A two-tailed non-parametric Wilcoxon matched pairs signed rank test was used to determine significant differences between two sets of paired values. Non-parametric Kruskal-Wallace test was used to determine significant differences within multiple data groups followed by Dunn's multiple comparison to identify differences between individual groups. A two-tailed non-parametric Spearman test was used to compute correlation coefficients between two variables. Raw data values were unadjusted for all statistical analyses. The threshold for significance was defined as $p < 0.05$. So as not to exclude datapoints from display that include a true value of 0, where data are displayed in figures on a $\log_{10}$ scale, IFNγ ELISpot response values of less than 1 SFU per $10^6$ cells were assigned a value of 1 and IFNγ flow cytometry responses of less than 0.01% of CD8 T-cells were assigned a value of 0.01.

## Ethics

Work was approved by the local ethics review boards, including the Rwanda National Ethics Committee, the Uganda Virus Research Institute Science and Ethics Committee (Currently the UVRI Research Ethics Committee) and the Uganda National Council of Science and Technology, the University of Cape Town Health Science Research and Ethics Committee, the University of Zambia Research Ethics Committee and the Emory University Institutional Review Board. Written informed consent was obtained from all participants.

## Results

### Polyclonal CD8 T-cell expansion

For the 13 subjects living with HIV, starting with 1 vial of cryopreserved PBMC (median 9.4, minimum 4.2 million PBMC), 7-day bi-specific antibody expansion resulted in median 121.4 million viable cells (Table 4). For 12 of these subjects, approximately 2/3 of the 7-day cells were removed, washed and rested for >22 hours. For subject 7 with the fewest expanded cells (38 million), all 7-day cells were washed and rested. Median viable cell numbers following resting were 98.5% of those removed on day 7. This provided sufficient cells in all 13 subjects for the first 3-D matrix peptide pool ELISpot requiring 38 to 76 million cells for 100,000 to 200,000 cells per ELISpot well in duplicate. The remaining 1/3 of day 7 cells (median 44.5 million) were cultured for a further 3 days and yielded median 97.7 million viable cells on day 11 following washing and >22 hours resting. These cells were tested in ELISpot with single peptides, selected based on responses to the first 3-D matrix ELISpot. CD8 T-cells were expanded from a second PBMC vial for single peptide ELISpot for subject 7 with insufficient cells expanded on day 7. For 7 HIV-1 seronegative subjects a median 12.1 million starting PBMC resulted in

**Table 4. CD8 T-cell expansion and number of PTE peptides and epitope regions recognised.**

| Subject | Viable cell numbers (millions) | | | | Number of recognised PTE peptides & epitope regions in HIV-1 proteins | | | | | | | | | |
| | Starting PBMC | Day 7 total cells | Day 7 cultured cells | Day 11 total cells | Gag | | Nef | | Pol | | Env | | Total | |
| | | | | | Peptide | Region | Peptide | Region | Peptide | Region | Peptide | Region | Peptide | Region |
| 1 | 12.3 | 311 | 83 | 248 | 0 | 0 | 4 | 2 | 1 | 1 | 0 | 0 | 5 | 3 |
| 2 | 7.1 | 85 | 19 | 63 | 5 | 3 | 6 | 3 | 1 | 1 | 2 | 2 | 14 | 9 |
| 3 | 9.4 | 121 | 29 | 85 | 1 | 1 | 13 | 2 | 14 | 9 | 0 | 0 | 28 | 12 |
| 4 | 12.0 | 190 | 63 | 110 | 19 | 8 | 3 | 2 | 8 | 4 | 3 | 2 | 33 | 16 |
| 5 | 9.7 | 176 | 97 | 163 | 9 | 5 | 14 | 5 | 29 | 21 | 13 | 10 | 65 | 41 |
| 6 | 5.7 | 63 | 16 | 40 | 7 | 2 | 19 | 6 | 6 | 3 | 6 | 5 | 38 | 16 |
| 7 | 4.5 | 38 | Not done | Not done | 1 | 1 | 2 | 1 | 26 | 9 | 1 | 1 | 30 | 12 |
| 8 | 6.3 | 98 | 25 | 85 | 29 | 11 | 5 | 2 | 12 | 7 | 2 | 2 | 48 | 22 |
| 9 | 9.0 | 234 | 117 | 180 | 22 | 8 | 14 | 4 | 17 | 8 | 5 | 4 | 58 | 24 |
| 10 | 11.6 | 181 | 101 | 187 | 9 | 6 | 6 | 2 | 36 | 25 | 8 | 3 | 59 | 36 |
| 11 | 4.2 | 83 | 23 | 50 | 8 | 5 | 0 | 0 | 8 | 4 | 2 | 2 | 18 | 11 |
| 12 | 9.6 | 93 | 22 | 71 | 3 | 3 | 11 | 5 | 14 | 9 | 1 | 1 | 29 | 18 |
| 13 | 9.5 | 180 | 60 | 120 | 13 | 8 | 6 | 3 | 7 | 5 | 11 | 4 | 37 | 20 |
| **Mean** | 8.5 | 142 | 54 | 117 | 9.7 | 4.7 | 7.9 | 2.8 | 13.8 | 8.2 | 4.2 | 2.8 | 35.5 | 18.5 |
| **Median** | 9.4 | 121 | 45 | 98 | 8 | 5 | 6 | 2 | 12 | 7 | 2 | 2 | 33 | 16 |
| **Min** | 4.2 | 38 | 16 | 40 | 0 | 0 | 0 | 0 | 1 | 1 | 0 | 0 | 5 | 3 |
| **Max** | 12.3 | 311 | 117 | 248 | 29 | 11 | 19 | 6 | 36 | 25 | 13 | 10 | 65 | 41 |

median 176.2 million cells after 7 days of expansion. Cells from these subjects were not assessed in day 11 single peptide ELISpot due to lack of any responses to matrix pools.

### PTE peptide IFNγ ELISpot responses

CD8 T-cells removed directly from bi-specific antibody/IL-2 culture without washing and resting are highly activated and result in high IFNγ ELISpot magnitudes without further stimulation. Across all 13 subjects living with HIV and 7 seronegative subjects, minimal release of IFNγ was detected in ELISpot assay of washed and rested expanded CD8 T-cells in the absence of further stimulation (maximum 7.5 SFU per million cells with mock stimulus). CD8 T-cells were functional in all subjects demonstrated by minimum PHA responses of 1235 SFU per million cells. No responses to the 1408 PTE peptides arranged in 174 3-D matrix pools were detected in CD8 T-cells expanded from 7 HIV-1 seronegative subjects (highest 15 SFU per million cells, mock-subtracted across 1218 matrix pool stimulations in duplicate). Responses to HIV-1 matrix PTE peptide pools were detected in all 13 subjects living with HIV. These were deconvoluted to narrow down the possible peptides recognised with actual recognised peptides identified in the second ELISpot on day 11.

A median of 197 (26 to 480) single PTE peptides were tested in the second ELISpot and a total of 462 PTE peptide responses (309 individual PTE peptides) were identified with a median of 33 peptide responses (5 to 65) per subject (Table 4 and S2 Table). The number of peptides recognised across the HIV-1 proteins tested was 126 Gag (27.3% of total), 103 Nef (22.3%), 54 Env (11.7%) and 179 Pol (38.9%). A single epitope region may account for more than one observed peptide response when variants of the same peptide are recognised or the amino acid sequences of two or more recognised peptides overlap [36]. Recognised PTE peptides were arranged by amino acid location relative to HXB2 protein sequences and were assigned to one epitope region where peptide sequences overlapped by 8 or more amino acids. As a result, the subjects' median 33 PTE peptides recognised could be explained by recognition of a median of at least 16 (3 to 41) epitope regions.

Alignment of peptide responses against a well-defined list of CD8 T-cell epitopes referred to as the A-list (HIV Sequence Compendium 2018) demonstrated that the responses detected using this expansion methodology are in line with responses reported in other studies. Of the 462 15mer peptides recognised, 114 (90.5%) of 126 Gag, 100 (97.1%) of 103 Nef, 40 (74.1%) of 54 Env and 130 (72.6%) of 179 Pol peptides overlapped by at least 8 amino acids with the HXB2 sequence location of at least one A-list epitope. Overall 384 (83.1%) of the 462 peptides recognised were located in previously defined optimal CD8 T-cell epitope regions (Fig 1).

Subject 8, tested at 48 months post estimated acquisition, was selected for further assessment of peptide recognition at an earlier time point based on availability of PBMC vials, a relatively high number of peptides recognised (48) and apparent control of HIV-1 replication in vivo evidenced by sustained low plasma viral loads (maximum 1035/mL) and stable CD4 counts (minimum 472/μL) at time points following 1 year of estimated date of acquisition. Table 5 demonstrates that CD8 T-cells from this subject recognised 32 PTE peptides at 30 months with 19 Gag, 9 Pol and 4 Nef peptides. All 32 peptides were again recognised at 48 months with an additional 16 peptides: 10 Gag, 1 Nef, 3 Pol and 2 Env. 10 were variants of peptides recognised at month 30 with 9 of these being Gag peptides. Env peptides were not recognised at month 30, with responses to 2 peptides apparent at month 48. As well as a broadening of the peptides recognised, the median response magnitudes for each peptide increased from 285 (interquartile range (IQR) 103–482) to 857 (IQR 501–1475) SFU per million cells for the 32 peptides recognised at both time points, with median magnitude of 487 (IQR 249–1091) for all peptides recognised at month 48.

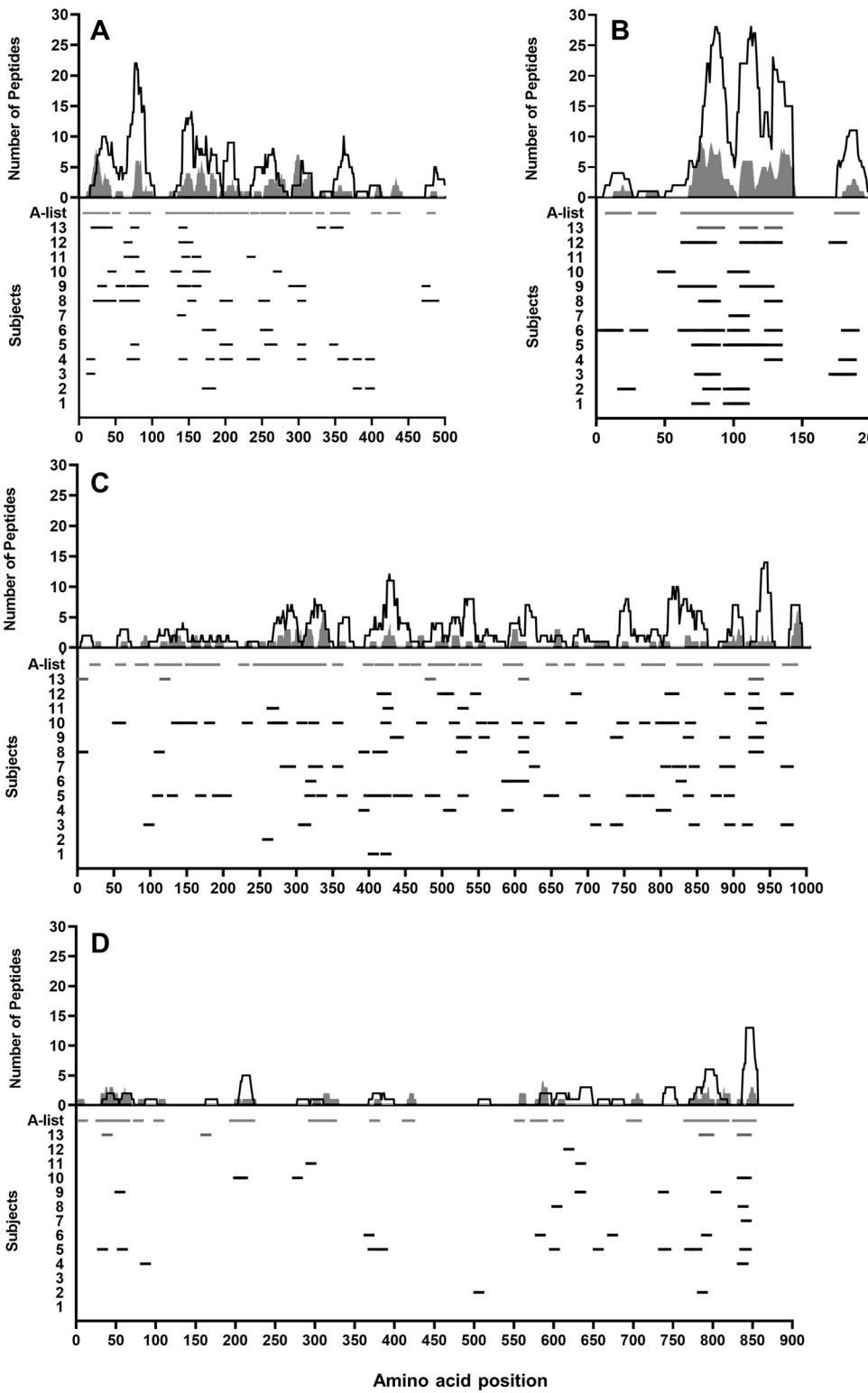

**Fig 1. Locations of PTE peptides recognised compared with A-list epitopes across HIV-1 proteins.** The x-axes represent HXB2 HIV-1 protein amino acid sequence position for each of the four represented proteins. The upper y-axes display the number of times an amino acid at each HXB2 amino acid sequence position is present in peptides recognised by the present study subjects (open area) compared with the A-list of known epitopes (shaded area). The lower y-axes display the study subjects and locations of peptide responses (black dash) as well as A-list epitope locations (shaded dash). (A) HIV-1 Gag protein (B) HIV-1 Nef protein (C) HIV-1 Pol protein (D) HIV-1 Env protein.

**Table 5. PTE peptide recognition by subject 8 at two post HIV-1 acquisition time points.**

| | Estimated months post acquisition | | 30 | | 48 | |
|---|---|---|---|---|---|---|
| HIV-1 Protein | HXB2[a] | Peptide number[b] | Peptide sequence | SFU[c] | Peptide sequence | SFU[c] |
| Gag | 25 | 175 | GKKHYMLKHLVWASR | 328 | GKKHYMLKHLVWASR | 1041 |
| | 25 | 303 | GKKHYMIKHLVWASR | 390 | GKKHYMIKHLVWASR | 1525 |
| | 32 | 20 | | | KHLVWASRELERFAL | 73 |
| | 37 | 68 | | | ASRELERFAVNPGLL | 327 |
| | 44 | 58 | | | FALNPGLLETSEGCR | 280 |
| | 45 | 167 | ALNPGLLETAEGCQQ | 120 | ALNPGLLETAEGCQQ | 714 |
| | 61 | 189 | LGQLQPALQTGSEEL | 68 | LGQLQPALQTGSEEL | 527 |
| | 62 | 317 | | | GQLHPSLQTGSEELK | 85 |
| | 64 | 118 | | | LQPSLQTGSEELRSL | 206 |
| | 67 | 182 | ALQTGTEELKSLYNT | 518 | ALKTGTEELRSLYNT | 133 |
| | 67 | 219 | ALQTGSEELRSLFNT | 135 | ALQTGTEELKSLYNT | 1094 |
| | 67 | 249 | | | ALQTGSEELRSLFNT | 425 |
| | 71 | 120 | GSEELRSLYNTVATL | 115 | GSEELRSLYNTVATL | 514 |
| | 71 | 123 | GSEELKSLYNTVATL | 343 | GSEELKSLYNTVATL | 687 |
| | 74 | 88 | | | ELRSLFNTVATLYCV | 367 |
| | 76 | 220 | KSLYNTVAVLYCVHQ | 105 | KSLYNTVAVLYCVHQ | 414 |
| | 76 | 289 | KSLFNTVATLYCVHA | 110 | KSLFNTVATLYCVHA | 545 |
| | 77 | 54 | SLYNTVATLYCVHQR | 175 | SLYNTVATLYCVHQR | 907 |
| | 154 | 75 | AWVKVVEEKGFNPEV | 38 | NAVKVVEEKAFSPEV | 153 |
| | 198 | 47 | MEMLKDTINEEAAEW | 95 | MEMLKDTINEEAAEW | 793 |
| | 198 | 80 | MHMLKETINEEAAEW | 98 | MHMLKETINEEAAEW | 807 |
| | 204 | 16 | | | TINEEAAEWDRLHPV | 433 |
| | 251 | 185 | | | TSNPPVPVGDIYKRW | 87 |
| | 255 | 53 | PIPVGDIYKRWIILG | 105 | PIPVGDIYKRWIILG | 306 |
| | 304 | 82 | LRAEQATQDVKNWMT | 615 | LRAEQATQDVKNWMT | 1080 |
| | 474 | 284 | | | PKQEQKDKELYPLAS | 38 |
| | 485 | 90 | PLTSLKSLFGSDPLS | 63 | PLTSLKSLFGSDPLS | 460 |
| | 486 | 208 | LTSLRSLFGSDPLSQ | 243 | LTSLRSLFGSDPLSQ | 1294 |
| | 486 | 217 | LTSLRSLFGNDPLSQ | 83 | LTSLRSLFGNDPLSQ | 527 |
| Nef | 81 | 29 | YKGAFDLSFFLKEKG | 338 | YKGAFDLSFFLKEKG | 285 |
| | 84 | 43 | | | AFDLGFFLKEKGGLE | 238 |
| | 129 | 10 | GPGVRYPLTFGWCFK | 673 | GPGVRYPLTFGWCFK | 2212 |
| | 129 | 27 | GPGIRYPLTFGWCYK | 430 | GPGIRYPLTFGWCYK | 1612 |
| | 129 | 31 | GPGTRFPLTFGWCFK | 1250 | GPGTRFPLTFGWCFK | 2146 |
| Pol | 7 | 260 | | | AFPQGEAREFPSEQT | 144 |
| | 112 | 285 | | | VKQYDQILIEICGKK | 44 |
| | 393 | 168 | KWTVQPIQLPEKDSW | 503 | KWTVQPIQLPEKDSW | 1458 |
| | 412 | 4 | IQKLVGKLNWASQIY | 525 | IQKLVGKLNWASQIY | 1044 |
| | 418 | 103 | KLNWASQIYPGIKVR | 75 | KLNWASQIYPGIKVR | 190 |
| | 418 | 136 | KLNWASQIYAGIKVK | 573 | KLNWASQIYAGIKVK | 1178 |
| | 527 | 304 | VQKIAMESIVIWGKT | 440 | VQKIAMESIVIWGKT | 1584 |
| | 527 | 477 | VQKVVMESIVIWGKA | 45 | VQKVVMESIVIWGKA | 317 |
| | 612 | 329 | | | YVTDRGRQKIVSLTE | 97 |
| | 928 | 151 | LQKQIIKIQNFRVYY | 580 | LQKQIIKIQNFRVYY | 2291 |
| | 933 | 301 | TKIQNFRVYYRDNRD | 378 | TKIQNFRVYYRDNRD | 1751 |
| | 934 | 42 | KIQNFRVYYRDSRDP | 475 | KIQNFRVYYRDSRDP | 2332 |

(*Continued*)

**Table 5.** (Continued)

| | | | Estimated months post acquisition | 30 | | 48 | |
|---|---|---|---|---|---|---|---|
| HIV-1 Protein | HXB2[a] | Peptide number[b] | | Peptide sequence | SFU[c] | Peptide sequence | SFU[c] |
| Env | 604 | 175 | | | | CTTAVPWNSSWSNRS | 60 |
| | 838 | 428 | | | | RAILHIPTRIRQGFE | 106 |

[a]Location of peptides' first amino acid relative to HIV-1 HXB2,

[b]PTE peptide number as supplied,

[c]Spot forming units (SFU) per million expanded CD8 T-cells.

### IFNγ responses of unexpanded and expanded CD8 T-cells

Thawing of 4 to 6 PBMC vials per subject living with HIV (n = 6) provided 16.4 to $37.5 \times 10^6$ PBMC after 23 hours rest. Negative selection led to 6.5 to $14.5 \times 10^6$ CD8 T-cells (median $9.9 \times 10^6$ cells or 36.9% of recovered PBMC), sufficient for ELISpot analyses of a limited number of CMVpp65 and 1st dimension matrix Nef peptide pools, rather than all 3 dimensions. IFNγ ELISpot responses (SFU/$10^6$ cells) to CMVpp65 or Nef matrix peptide pools were similar between 10-day expanded CD8 T-cells and CD8 T-cells isolated directly from PBMC (medians 158 and 130 SFU/$10^6$ cells, respectively, p = 0.427) (Fig 2A) with a significant and strong positive correlation (r = 0.814, p <0.001) (Fig 2B). Similar findings were apparent in a separate study group when PBMC and expanded CD8 T-cells were assessed by intra-cellular cytokine flow cytometry following stimulation with CMVpp65 or HIV-1 Gag, Nef, Pol and Env PTE peptide pools. %IFNγ+ CD8 T-cells were similar between expanded CD8 T-cells and CD8 T-cells isolated directly from PBMC (medians 0.417 and 0.347%, respectively, p = 0.314) (Fig 3A) with a significant positive correlation (r = 0.691, p <0.001) (Fig 3B).

### Expanded/Cryopreserved CD8 T-cells

Day 10 expanded and rested CD8 T-cells were cryopreserved in liquid nitrogen vapour. There was no significant difference in ELISpot responses of these cells to HIV-1 Gag, Nef, Pol or Env peptide pools following thaw and overnight rest compared with expanded CD8 T-cells tested prior to cryopreservation (p = 0.854) with a significant positive correlation (r = 0.853, p <0.001) (S1 Fig).

### LucR IMC replication in expanded CD4 T-cells and inhibition by autologous CD8 T-cells

All 10 IMC replicated in expanded CD4 T-cells from all subjects with $\log_{10}$ RLU values between 3.5 and 5.6 over media only background. Inhibition of IMC replication in CD4 T-cells above the positive cut-off of 0.8 $\log_{10}$ reduction in RLU values was evident following co-culture with CD8 T-cells from all subjects with median of 7 IMC inhibited (range 4 to 10) (Fig 4A and S3 Table). CD8 T-cells from two subjects living with HIV-1 clade C (6 and 9) inhibited all ten IMC. Arranging data by both subject clade and test HIV-1 IMC clade indicated that for subjects living with the same HIV-1 clade there was no significant difference in inhibition of test HIV-1 IMC of different clades (within clade A subjects: p = 0.167, clade C: p = 0.920 or clade D: p = 0.182) (Fig 4B). Grouping all inhibition values by subject HIV-1 clade demonstrated that 66%, 78.3% and 40% of IMC were inhibited above the 0.8 $\log_{10}$ positive inhibition value by subjects living with HIV-1 clade A, clade C or clade D, respectively. $\log_{10}$ Inhibition values were significantly higher for subjects living with clade A (median 0.93, p = 0.027) and C

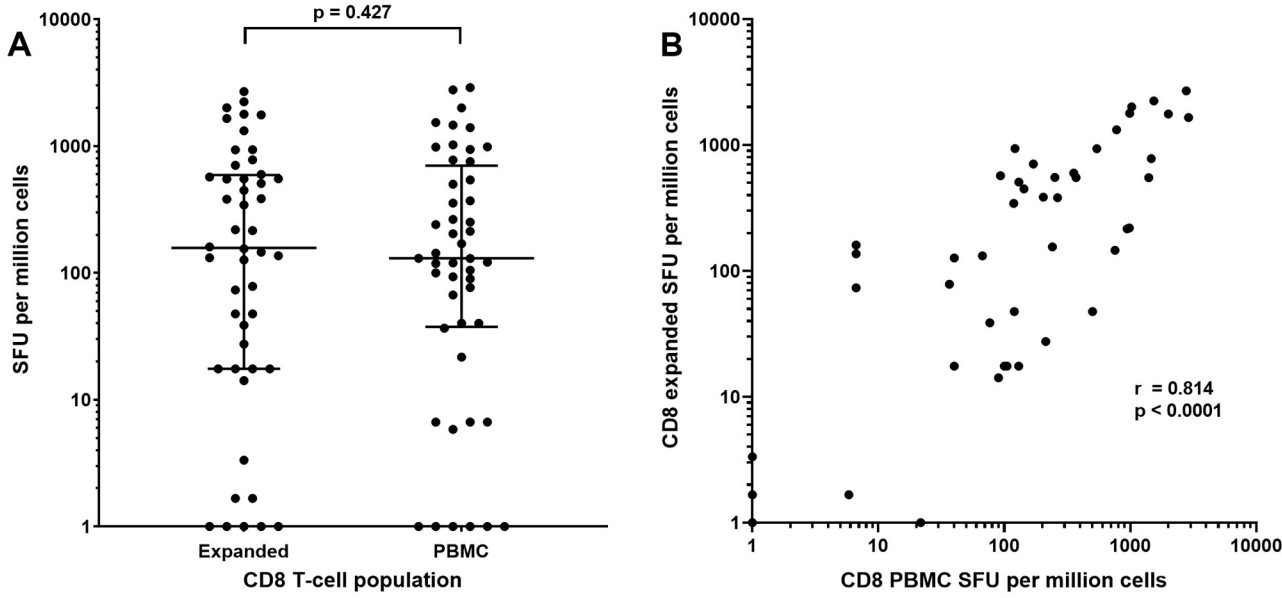

**Fig 2. IFNγ ELISpot responses to CMVpp65 or HIV-1 Nef peptide pools in expanded CD8 T-cells or CD8 T-cells isolated from PBMC.** A) SFU per million cells with medians (bar) and interquartile ranges and B) correlation between data sets with a non-parametric Spearman test used to compute the correlation coefficient (r) between the two the datasets. For $\log_{10}$ scale display only, values of <1 SFU were assigned a value of 1. Statistical analyses were conducted using unaltered data.

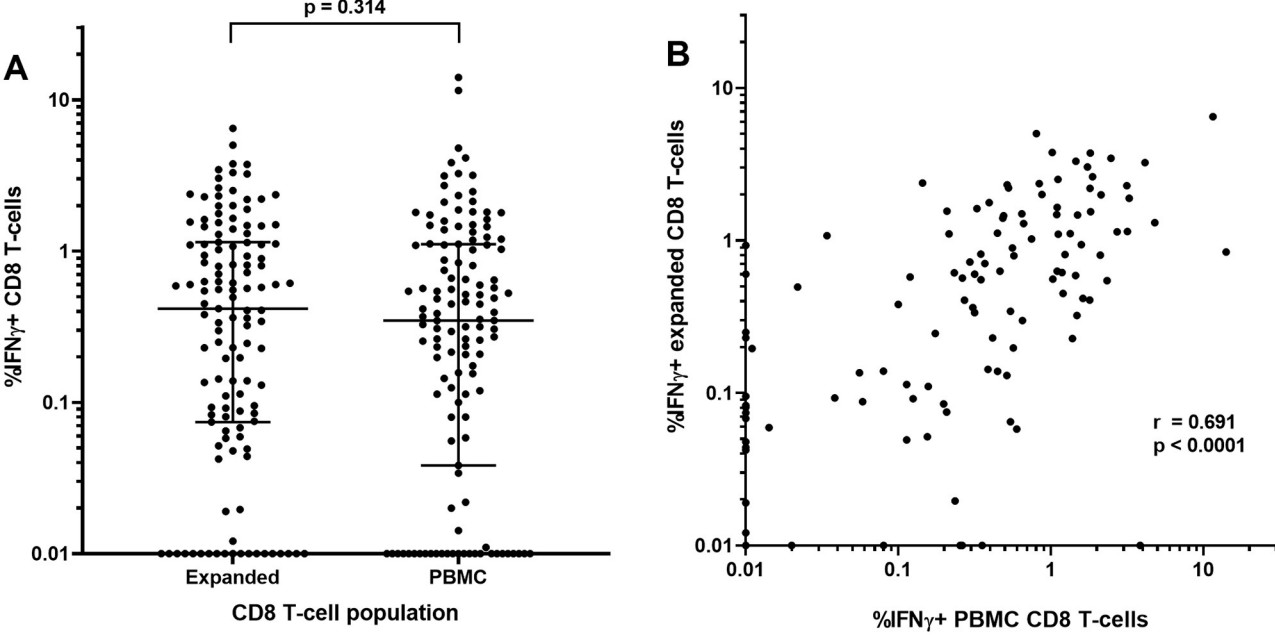

**Fig 3. Flow cytometric analyses of IFNγ+ events of CD3+CD8+ T-cells stimulated with CMVpp65 or HIV-1 PTE peptide pools in expanded CD8 T-cells or PBMC.** A) percent IFNγ+ events with medians (bar) and interquartile ranges and B) correlation between datasets with a non-parametric Spearman test used to compute the correlation coefficient (r) between the two datasets. For $\log_{10}$ scale display only, values of <0.01 SFU were assigned a value of 0.01. Statistical analyses were conducted using unaltered data.

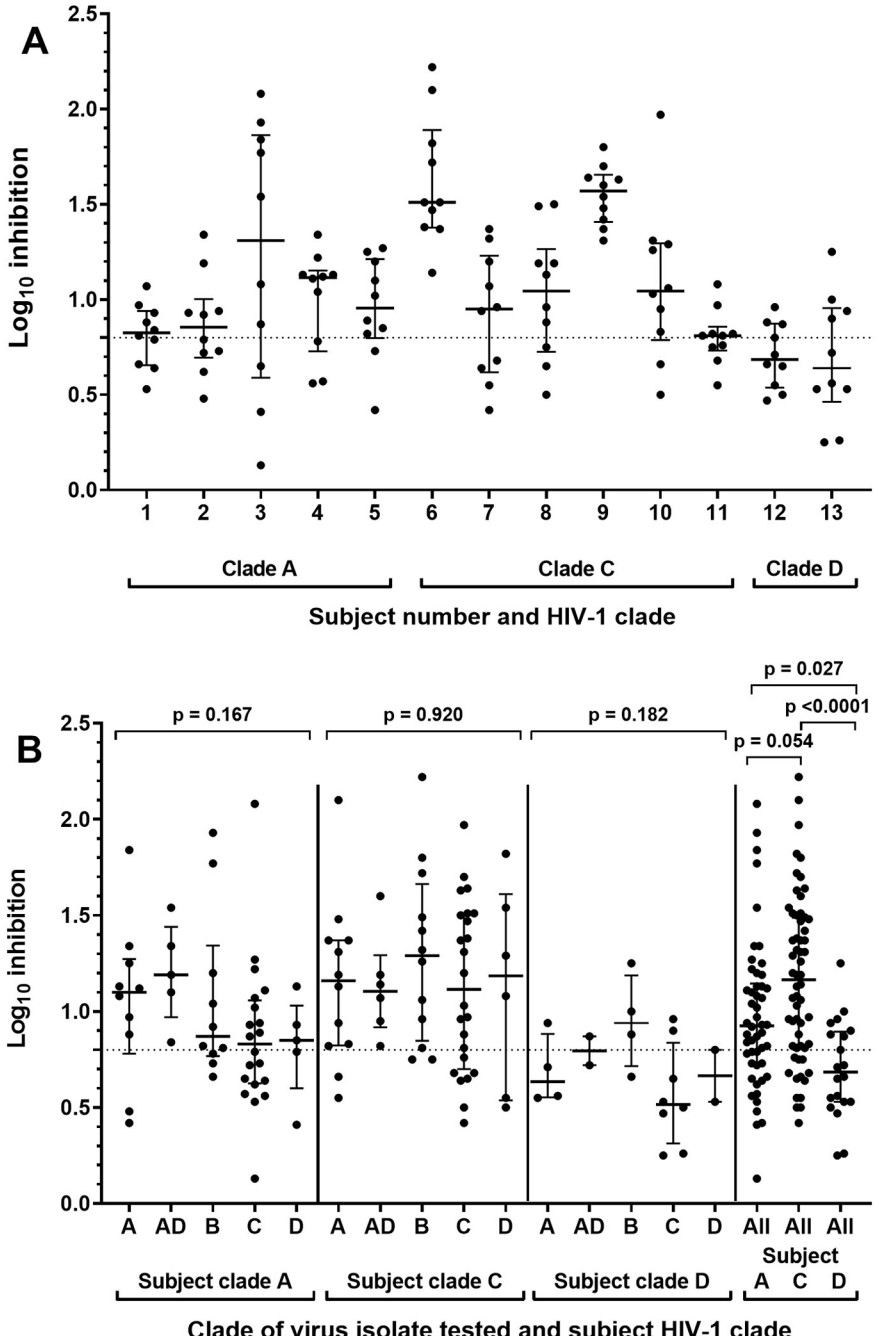

**Fig 4. CD8 T-cell mediated inhibition of HIV-1 replication in autologous CD4 T-cells.** A) Inhibition for each subject living with HIV, arranged by subject HIV-1 clade, B) Inhibition arranged by subject HIV-1 clade and test HIV-1 LucR IMC clade. Each of the 3 left panels, inhibition values for test viruses of the same clade are grouped together for subjects living with the same HIV-1 clade (A, C or D clades). In the right panel, inhibition values for all test viruses are grouped by subject HIV-1 clade. Bars represent the medians with interquartile ranges, the dotted line represents the assay positivity value (>0.8 log10).

(median 1.17, p < 0.001) compared with clade D (median 0.69). Inhibition values for subjects living with HIV-1 clade C were marginally higher than clade A, though not significantly (p = 0.054) (Fig 4B).

## Peptide recognition and HIV-1 inhibition breadth

There was a significant positive correlation between the number of HIV-1 isolates inhibited and the number of peptides recognised by subjects, including variants of the same peptide (r = 0.664, p = 0.016) (Fig 5A). There was also a positive but not significant correlation between the breadth of inhibition and the minimum number of epitope regions recognised by CD8 T-cells (r = 0.483, p = 0.096) (Fig 5B). Positive but not significant correlations were also apparent between inhibition breadth and the number of Gag (r = 0.355, p = 0.231), Nef (r = 0.503, p = 0.082), Env (r = 0.378, p = 0.201) and Pol (r = 0.420, p = 0.53) peptides recognised.

## Phylogenetic analysis of viruses in the VIA panel

The IMC were derived from the major nodes in HIV-1 subtypes A, B, C and D including one subtype A1/D recombinant. As most subjects from which these sequences were derived acquired HIV-1 subtype A, C, D or unique recombinant forms of these subtypes, there was insufficient representation of subtype B sequences to evaluate the breadth of representation of the subtype B IMC. The selected IMC also represent a diversity of distances from the protocol C consensus sequence and are therefore representative of the transmitted/founder (T/F) variants in this cohort (S2 Fig).

## Assessment of HIV-1 replication in expanded CD8 T-cells from subjects living with HIV

Day 13 cultures of expanded CD4 T-cells infected in vitro with HIV-1 isolates demonstrated HIV-1 replication with release of HIV-1 p24 Gag protein into culture supernatants of between $1.3 \times 10^5$ and $3.2 \times 10^5$ pg/mL. Minimal release of p24 Gag protein that would be derived from subjects' own acquired HIV-1 was detected in day 10 expanded CD8 T-cells from 71 ARV-naïve subjects living with HIV, where no values were above the assay positive cut-off value of 5.8pg/mL (mean $OD_{490}$ of media only wells + 0.05). 12 of 71 supernatants did record $OD_{490}$ values below the supplier assay positive cut off value but above the media only wells, which would be equivalent to p24 Gag concentrations between 1.1 and 2.8pg/mL (S3 Fig).

## Discussion

The current understanding of the role of T-cells in HIV-1 pathogenesis is based on many separate studies. Due to limits in PBMC sampling, each study typically assesses a limited scope of T-cell functions, often with diverse subject groups between studies. Whilst these studies have yielded valuable data and are the basis of our knowledge of HIV-1 pathogenesis, the ability to comprehensively assess and link multiple HIV-1 specific T-cell functions from the same, often limited, study sample set would further enhance our understanding of the immune response to HIV-1 infection and potentially assist in the design and evaluation of candidate HIV-1 vaccines. The present study links two fundamental aspects of CD8 T-cell biology: identification of HIV-1 peptides recognised by CD8 T-cells and the ultimate outcome of this peptide recognition: the extent of CD8 T-cell mediated inhibition of replication of a diverse panel of HIV-1 isolates in autologous CD4 T-cells.

Of all T-cell functions that could be assessed, the use of ELISpot or other assays to map the recognition of individual peptides would require the most cells due to the number peptides

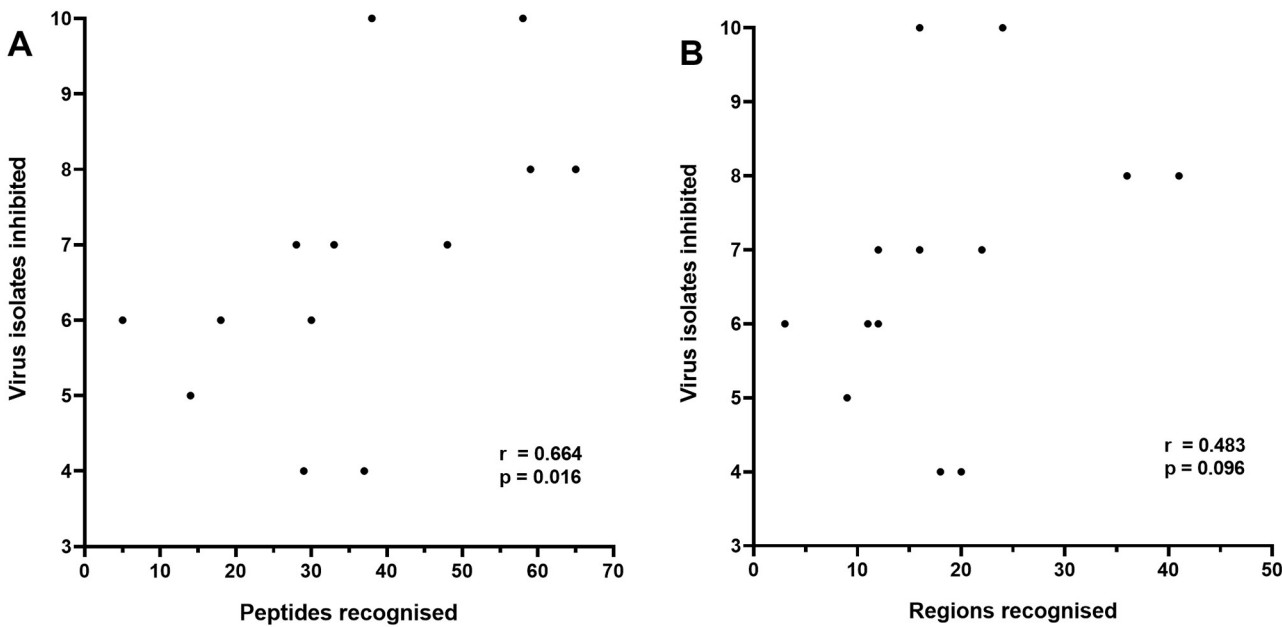

**Fig 5. Correlations between the number of HIV-1 isolates inhibited and A) the total number of PTE peptides recognised and B) the total number of PTE epitope regions recognised.** A non-parametric Spearman test was used to compute the correlation coefficients (r) between the datasets.

needed to match extensive HIV-1 sequence variability. One approach for comprehensive assessment of T-cell epitope recognition is the PTE peptide approach exemplified by a peptide set covering the majority of the HIV-1 proteome designed to maximise coverage of 9 amino acid sequences potentially recognised as peptide epitopes by CD8 T-cells with the minimum number of 15mer peptides. This approach results in an extensive 1408 peptide set which has not been widely employed by other groups. One study [37] utilised PBMC to assess breadth of CD8 T cell peptide recognition using a three step ELISpot testing regime, first with HIV-1 PTE peptide master pools of up to 100 peptides followed by peptides matrices for responding master pools and then single peptide testing. Such an approach was feasible in this instance as subjects had recently acquired HIV-1 at a median of 69 days post infection with a median of 7 (range 3 to 13) epitopes identified per subject. The few epitopes recognised would result in a minority of master peptide pools then requiring assessment with the more extensive peptide matrices and single peptides in ELISpot assay. In the present study of clinically latent subjects, using the above approach the distribution of the higher number of peptide responses would require many more cells with around a median of 10 peptide master pools responding, each requiring assessment with peptide matrix pools followed by single peptide ELISpot.

Although refinements to the 3-D matrices used in the present study may reduce the required total cell number, whatever approach is taken will still require many hundreds of millions of cells. For a reliable approach using 200 000 cells per ELISpot well plated in duplicate, rather than fewer cells in single wells, the coverage of all 1408 PTE peptides requires 76 million cells for the 3-D matrix ELISpot and then a median of 84 million cells for confirmation of single peptide ELISpot responses. Combining this with multi-parameter flow-cytometric analyses of T-cell responses to several HIV-1 and control peptide pool stimuli would typically require around an additional 8 million cells (1 million per stimulation). In addition, the LucR VIA method described in this report would require a further 2 million CD8 T-cells which if not derived by expansion would require direct isolation from approximately 10 million PBMC

along with 4 million CD4 T-cells or possibly CD8-depleted PBMC. Such combined mapping, flow cytometric and viral inhibition analyses would require on average around 180 million PBMC. The logistical complications and multiple laboratory operator requirement of such an approach would be facilitated by the use of frozen PBMC where typically around 65% of PBMC frozen may be viably recovered, meaning around 280 million PBMC would be required on average prior to cryopreservation, with more required in many subjects, typically requiring around 300mL blood. 300mL of blood to capture all these immunological assessments may be possible in a limited number of willing and consented volunteers living with clinically latent HIV, but is unlikely to be available for a larger number of subjects at different HIV stages. Such studies would also quite rightly be subject to critical review for local ethical approval. In most cases the only way to perform such multiple immunological assessments in tandem on the same sample set is to polyclonally expand T-cells.

In the present study, median 9.4 million PBMC recovered per frozen vial of PBMC yielded 38 to 311 million CD8 T-cells by day 7 of polyclonal expansion for the first matrix pool ELISpot. There was a further approximate doubling of viable cell numbers in cultures continued from day 7 to 11 for single peptide ELISpot. Any remaining or even all expanded CD8 T-cells can be cryopreserved for additional or later analyses as we found no significant differences in the CD8 T-cell ELISpot responses using expanded cells that were used immediately or cryopreserved. The majority (83.1%) of HIV-1 peptides identified as being recognised by expanded CD8 T-cells correspond with well described CD8 T-cell epitopes reported in other studies [29, 38] indicating that the expansion procedure does not result in CD8 T-cells with atypical specificities. Although not a study utilising the PTE peptide approach, IFNγ ELISpot assay of PBMC was applied to assess recognition of 504 overlapping peptides representing all expressed clade B HIV-1 proteins within subjects presumed to be living with clade B HIV-1 [36]. 95% of responses were mediated by CD8 T-cells with anti-retroviral naïve subjects with clinical latency recognising a median of 18.5 (2–42) epitope regions with 75% of these responses targeting Nef, Gag or Pol epitopes. Although responses to HIV-1 accessory proteins were not assessed, the median of 16 (3–41) epitope regions recognised in the present study is comparable. In addition to assessing epitope regions targeted, the PTE approach has the advantage of also assessing responses to peptide variants within epitope regions. The peptides recognised by expanded CD8 T-cells were assessed in one subject with low plasma viral loads at two clinically latent time points: 30 and 48 months post-estimated acquisition. All 32 peptides identified at the earlier time point were also recognised at the later time point. An additional 16 peptides recognised at the later time point consisted of 10 variants of earlier time point peptides and 6 new specificities. This finding may be explained by existing CD8 T-cell epitope specificities are retained along with evolution in existing epitope specificities and responses to new epitopes as HIV-1 sequences vary during ongoing chronic viral replication [7, 25].

Previous studies from our group applied a version of the viral inhibition assay to studies of HIV-1 pathogenesis [17] and clinical trial of HIV-1 vaccine candidates [18, 19] where HIV-1 replication in CD4 T-cells was assessed by measuring HIV-1 p24 protein release into culture supernatants. CD8 T-cell mediated inhibition of HIV-1 replication was demonstrated by reduced p24 release, with such inhibition shown to be MHCI dependent. Development of a less labour-intensive modified VIA employs HIV-1 IMC engineered with a luciferase gene to report the extent of HIV-1 replication in culture [20]. The present report extended this approach with a HIV-1 LucR IMC panel representing multiple HIV-1 clades [21]. 9 of the 10 HIV-1 IMC were obtained from subjects with recent HIV acquisition (Table 1) and such HIV-1 sequences would represent circulating and transmitted/founder viruses that any effective HIV-1 vaccine would presumably have to elicit an immune response to. A tenth LucR IMC was derived from the common HIV-1 laboratory strain NL4-3 and represents a useful control

for which various matched laboratory assay reagents are available, such as HIV-1 peptides and proteins.

Whilst CD8 T-cells from all subjects tested inhibited at least 4 LucR IMC with a median of 7, two subjects, both living with HIV-1 clade C, inhibited all 10 viruses, warranting a closer examination of the potential influence of subject HIV-1 clade on inhibition values. Grouping all 10 test viruses together, $\log_{10}$ inhibition values were significantly higher for subjects living with clade A (p = 0.027) and C (p < 0.001) compared with clade D (p < 0.001) with a trend towards higher values with subjects living with HIV-1 clade C compared with clade A (p = 0.054) (Fig 4B). The reason for more efficient inhibition observed in subjects living with HIV-1 clade C does not appear to be due to a higher proportion of LucR IMC test viruses being of clade C (4 of 10) as there were no significant differences in median $\log_{10}$ inhibition values for the 5 different clades of test viruses in subjects living with the same HIV-1 clade, including clade C. Different reports have suggested that subjects living with clade C HIV-1 may have a faster or slower disease progression or that there is no effect [39]. The potential effect of HIV-1 clade acquisition on cross-clade inhibition breadth warrants further study in a larger group of subjects.

One key finding of the present report is that the breadth of HIV-1 inhibition in terms of the number of IMC inhibited was significantly (p = 0.016) and positively correlated (r = 0.664) with the number of PTE peptides recognised. There was also a trend towards significance in the correlation between HIV-1 inhibition breadth and the number of HIV-1 epitope regions recognised. Therefore, as subjects' CD8 T-cells recognise more epitope regions and in particular more epitope variants within these regions, these CD8 T-cells can recognise and inhibit a wider variety of HIV-1 isolates. Such findings add support to the notion that vaccine candidates designed to elicit CD8 T-cells may have more chance of success where broad epitope recognition is induced. The simultaneous assessment of multiple functions described in this report can be employed to further characterise the role of CD8 T-cells in HIV-1 pathogenesis to inform on vaccine design and also to then assess the nature of CD8 T-cell responses elicited by vaccine candidates, prioritising more promising candidates for further development and clinical trial.

As expected, there was evidence of only minimal productive HIV-1 replication within cultures of expanded CD8 T-cells from subjects living with HIV assessed by culture supernatant p24 gag protein content, with all values below the supplier assay positive cut-off value. ARV-naïve subjects living with HIV with a wide range of plasma viral loads were selected as subjects who may have a greater potential for replication of their own acquired HIV-1 in vitro. This finding provides information relevant to the laboratory safety aspects of the culture, expansion and ELISpot analysis of CD8 T-cells from subjects living with HIV.

A valid criticism of the expansion approach described in this report is that HIV-1 specific CD8 T-cells polyclonally expanded in vitro with CD3/CD4 bi-specific antibody may have very different phenotypes and functions to that of ex vivo CD8 T-cells within PBMC. In terms of CD8 T-cell phenotype, expanded HIV-1 specific CD8 T-cells have a memory or effector/memory phenotype [19], a mix of phenotypes reported for HIV-1 specific ex vivo CD8 T-cells in other studies [40]. In terms of functional responses, equivalent inhibition of HIV-1 replication mediated by expanded CD8 T-cells or CD8 T-cells isolated directly from PBMC has been demonstrated [19]. A recent report from our group demonstrated that polyclonally-expanded and directly-isolated CD8 T-cells displayed similar antigen sensitivities in IFNγ ELISpot assay [24]. The present study extended and confirmed these findings and demonstrated a significant and strong positive correlation between expanded and PBMC-derived CD8 T-cell IFNγ responses to HIV-1 and CMVpp65 peptide pools, assessed in both ELISpot and flow cytometric analyses. Taken together, these results demonstrate that CD3/CD4 polyclonal expansion yields CD8 T-

cells with comparable phenotypes and functions to that of ex vivo CD8 T-cells within PBMC, with the expansion approach often being the only possible means of assessing multiple CD8 T-cell functions simultaneously within typical subject blood-sampling limits.

In conclusion, the expansion approach allows detailed assessment of multiple CD8 T-cell functions from a limited sample volume, in this case both mapping of individual peptides recognised by CD8 T-cells in comprehensive PTE peptide sets and assessment of CD8 T-cell mediated inhibition of replication of a diverse and cross-clade panel of HIV-1 isolates. This approach demonstrated a positive correlation between CD8 T-cell mediated breadth of HIV-1 inhibition and peptide recognition that would support a theory that an effective HIV-1 vaccine eliciting a cellular response would need to elicit T-cells recognising a wide breadth of HIV-1 epitopes. Studies are currently underway within our group, where a larger number of subjects from the prospective incident HIV-1 acquisition cohort have been carefully selected [30] and matched to represent a spectrum of HIV-1 progression rates, clade of HIV-1, geographic location and gender with samples assessed at both the acute and clinically latent time points. Overall, 1 to 2 vials of frozen PBMC (10 million PBMC frozen per vial) would yield enough expanded CD8 T-cells to allow concurrent mapping of peptides recognized, flow cytometric analyses of intracellular cytokine responses to several HIV-1 and control stimuli and assessment of CD8 T-cell mediated inhibition of replication of a diverse panel of HIV-1 isolates.

An ability to assess breath of CD8 T-cell function against a panel of HIV-1 isolates and mapping of peptide specificities associated with anti-viral function, has not been possible to date. However, such a strategy was pivotal to the discovery of broadly neutralizing HIV-1 antibodies and determining antibody epitope specificities thereby enabling rational immunogen design and vaccine development [41–43]. The present approach provides tools to better characterise efficacious CD8 T-cells thereby facilitating rational design and evaluation of future candidate T-cell immunogens.

## Supporting information

**S1 Fig. Correlation between magnitudes (SFU per million cells) of IFNγ ELISpot responses to HIV-1 PTE peptide pools for expanded CD8 T-cells or expanded and cryopreserved CD8 T-cells.** A non-parametric Spearman test was used to compute the correlation coefficients (r) between the datasets.
(TIF)

**S2 Fig. Phylogenetic analysis of 201 HIV-1 full-length proteome sequences derived from IAVI protocol C.** Single genome nucleotide sequences for each viral gene (*gag*, *pol*, *vif*, *vpr*, *vpu*, *tat*, *rev*, *env*, *& nef*) were translated to amino acids, concatenated, aligned with LANL consensus/ancestral sequences and a maximum likelihood tree generated. Transmitted/founder sequences of subtype A are shown in red, subtype B in blue, subtype C in purple, subtype D in green and unique recombinant forms in grey. Infectious molecular clones in the LucR VIA are highlighted with a pink circle. Subtype references are highlighted with a black circle.
(TIF)

**S3 Fig. HIV-1 p24 gag protein content of culture supernatants.** 7 day expanded CD4 T-cells from a HIV-1 uninfected subject sampled at 13 days after in vitro infection with one of three HIV-1 isolates at an MOI of 0.01: 247Fv2 (square), IIIB (triangle) and CH077 (diamond) and 10 day expanded CD8 T-cells from 71 ARV-naïve subjects living with HIV without further HIV-1 infection in vitro (circles). Dotted line represents the assay positive cut-off value (mean

of replicate media only background wells $OD_{490}$ + 0.05 = 5.8pg/mL) as recommended by the ELISA supplier's instructions.
(TIF)

**S1 Table. Commercial reagents used for flow cytometry.** Cells were stained in 100μL staining volume at the dilution specified.
(DOCX)

**S2 Table. PTE peptides recognised by CD8 T-cells from study subjects and IFNγ ELISpot response magnitudes.**
(DOCX)

**S3 Table. CD8 T-cell mediated inhibition of HIV-1 replication.** Inhibition was determined by the $log_{10}$ reduction in relative light units of cultures of CD4 T-cells infected with one of ten luciferase gene engineered HIV-1 infectious molecular clones (IMC) and co-cultured with autologous CD8 T-cells compared with cultures of HIV-1 infected CD4 T-cells alone.
(DOCX)

## Acknowledgments

We thank the IAVI protocol C investigators who are: Eduard J. Sanders, Centre for Geographic Medicine—Coast/KEMRI, Kenya and University of Oxford, Oxford, United Kingdom. Omu Anzala, KAVI-Institute of Clinical Research, Nairobi, Kenya. Anatoli Kamali, IAVI, Nairobi, Kenya. Etienne Karita, Center for Family Health Research, Kigali, Rwanda. William Kilembe, Mubiana Inambao and Shabir Lakhi, Center for Family Health Research, Lusaka & Ndola, Zambia. Susan Allen and Eric Hunter, Emory University, Georgia, United States of America. Vinodh Edward, The Aurum Institute, Johannesburg and Rustenburg, South Africa. Pat Fast, IAVI, New York, United States of America. Matt A. Price, IAVI, New York and Department of Epidemiology and Biostatistics, University of California San Francisco, United States of America. Jill Gilmour, IAVI Human Immunology Laboratory, Imperial College, London, United Kingdom. Jianming Tang, School of Medicine, University of Alabama, United States of America. Fran Priddy, IAVI, New York, United States of America. Mary H. Latka, The Aurum Institute, South Africa. Linda-Gail Bekker, Desmond Tutu Health Foundation, University of Cape Town, Cape Town, South Africa. Pontiano Kaleebu, Medical Research Council, Uganda Virus Research Institute and London School of Hygiene and Tropical Medicine Uganda Research Unit (MULS), Entebbe & Masaka, Uganda. The investigators can be contacted via Matt A. Price (email MPrice@iavi.org).

## Author Contributions

**Conceptualization:** Peter Hayes, Deborah King, Julia Makinde, Eric Hunter, Jill Gilmour.

**Data curation:** Peter Hayes, Lucas Black, Gladys Macharia, Matt Price.

**Formal analysis:** Peter Hayes, Natalia Fernandez, Lucas Black, Gladys Macharia.

**Funding acquisition:** Jill Gilmour.

**Investigation:** Peter Hayes, Natalia Fernandez, Jama Dalel, Lucas Black, Claire Streatfield, Vanaja Kakarla.

**Methodology:** Peter Hayes, Natalia Fernandez, Christina Ochsenbauer, Jama Dalel, Jonathan Hare, Deborah King, Lucas Black.

**Project administration:** Peter Hayes, Deborah King.

**Resources:** Peter Hayes, Natalia Fernandez, Christina Ochsenbauer, Claire Streatfield, Vanaja Kakarla.

**Supervision:** Peter Hayes, Deborah King, Julia Makinde, Eric Hunter, Jill Gilmour.

**Validation:** Peter Hayes, Natalia Fernandez, Lucas Black.

**Visualization:** Peter Hayes, Natalia Fernandez, Jama Dalel, Lucas Black, Gladys Macharia.

**Writing – original draft:** Peter Hayes, Natalia Fernandez, Jama Dalel.

**Writing – review & editing:** Peter Hayes, Natalia Fernandez, Christina Ochsenbauer, Jama Dalel, Jonathan Hare, Deborah King, Lucas Black, Claire Streatfield, Vanaja Kakarla, Gladys Macharia, Julia Makinde, Matt Price, Eric Hunter, Jill Gilmour.

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
