## [Decision Letter · Decision Letter 0]

4 Oct 2021

PONE-D-21-25896Breadth of CD8 T-cell mediated inhibition of replication of diverse HIV-1 transmitted-founder isolates correlates with the breadth of recognition within a comprehensive HIV-1 Gag, Nef, Env and Pol potential T-cell epitope (PTE) peptide setPLOS ONE

Dear Dr. Hayes,

Thank you for submitting your manuscript to PLOS ONE. After careful consideration, we feel that it has merit but does not fully meet PLOS ONE’s publication criteria as it currently stands. Therefore, we invite you to submit a revised version of the manuscript that addresses the points raised during the review process. I agree with the reviewers' comments, and please use them to improve your manuscript.

We look forward to receiving your revised manuscript.

Kind regards,

Douglas F. Nixon, M.D., Ph.D.

Academic Editor

PLOS ONE

4. Please list the individual authors and affiliations within the group "The IAVI protocol C investigators" in the acknowledgments section of your manuscript. Please also indicate clearly a lead author for this group along with a contact email address.

Additional Editor Comments (if provided):

Reviewers' comments:

Reviewer's Responses to Questions

**Comments to the Author**

1. Is the manuscript technically sound, and do the data support the conclusions?

Reviewer #1: Yes

2. Has the statistical analysis been performed appropriately and rigorously? 

Reviewer #1: Yes

3. Have the authors made all data underlying the findings in their manuscript fully available?

Reviewer #1: Yes

4. Is the manuscript presented in an intelligible fashion and written in standard English?

Reviewer #1: Yes

5. Review Comments to the Author

Reviewer #1: Characterizing HIV-specific CD8+ T cell responses is important to understand the elicited antiviral response and can help in the generation of a vaccine. A major issue is that large quantities of cells are needed to perform an extensive characterization of the CD8s. The manuscript by Hayes and colleagues describes a method to overcome this limitation. In their study, they test if polyclonal expansion of CD8s yields the same response to HIV peptides as compared to unexpanded CD8s (directly from PBMCs). They find that the expanded CD8 T cells perform equally well as unexpanded cells to various HIV peptides and can also be cryopreserved, this is very convenient for logistic reasons. Overall the manuscript describes a great technical development that makes it possible to investigate the breadth of HIV-specific CD8 T cell responses using peptides and ELISpot and have enough cells to perform a virus inhibition assay, in which one can test if the CD8s inhibit HIV replication in allogeneic CD4s.

The manuscript is well written and organized, the data is presented clearly and the methodology is appropriately described. Overall, I have only a few minor comments that I hope can be of some use for this already nice manuscript.

I would like to encourage the authors to consider using less stigmatizing vocabulary. For example by using “person living with HIV (“PLHIV or PLWH)” instead of HIV-infected. For more information: http://www.hiveonline.org/wp-content/uploads/2016/01/Anti-StigmaSign-Onletter1.pdf

https://www.thewellproject.org/hiv-information/why-language-matters-facing-hiv-stigma-our-own-words.

Page 4, line 103. Typo in “where (should be were)

Page 5, line 112. “Subjects were draws from a large prospective cohort…” I think randomly selected would be more appropriate for the patient selection. (Or if they were selected based on specific criteria, then these criteria should be listed).

Page 9, line 195 and fig 2 on page 20: It would help the reader if it was written more explicitly that the CMV and HIV peptides were added together to the CD8s. Now it is written as xxx and xxx were added, but this can be interpreted as being added to two different wells. Also, please add the final concentration of peptides used (is this similar to the 1ug/mL for the PTE peptides?): for the CMV+Nef and CMV+HIVpep in the flow exp.

Page 20, line 395. SFU is explained in the legend of table 5 on page 19, but I think it would help the reader to also explain the abbreviation when first mentioned in the main text (but perhaps I missed this).

Fig 4. Typo on the x-axis legend: testred = tested

6. PLOS authors have the option to publish the peer review history of their article (what does this mean?). If published, this will include your full peer review and any attached files.

Reviewer #1: **Yes: **Renee M. van der Sluis

---

## [Author Response · Author response to Decision Letter 0]

1 Nov 2021

The specific reviewer and editor comments are detailed in the "response to reviewers" letter. I copy the relevant sections of this letter here:-

Journal requirements

1) I have checked and have ensured that the manuscript meets PLOS ONE’s style and file naming requirements.

2) I have revised the funding information:-

This work was made possible by IAVI, which was supported by funding from many donors, including the generous support of the American people through the US Agency for International Development award number AID-OAA-A-16-00032, the Bill and Melinda Gates Foundation, the Ministry of Foreign Affairs of Denmark, Irish Aid, the Ministry of Finance of Japan in partnership with The World Bank, the Ministry of Foreign Affairs of the Netherlands, the Norwegian Agency for Development Cooperation and the United Kingdom Department for International Development. The full list of IAVI donors is available at: http://www.iavi.org. The funding donors had no role in the study design, data collection and analysis, decision to publish, or preparation of the manuscript.

3) I confirm the ethics statement only appears in the Methods section of the manuscript.

4) The list of IAVI protocol C investigators and contact details are provided in the acknowledgements section. There is not one single lead author but the group can be contacted via Matt A. Price (email MPrice@iavi.org).

The reference list has been reviewed, remains unchanged and none of the cited papers have been retracted.

Reviewer’s comments

In comments 1 to 4, the reviewer stated the manuscript complies with the requirements.

Comment 5: I thank the reviewer for highlighting the need to use less stigmatizing vocabulary. I have reviewed the information provided and have revised the manuscript accordingly. “HIV-1 infected subjects” and similar terminology has been replaced with “subjects living with HIV”. “HIV-1 infection” has been replaced with “acquisition of HIV”. Such changes can be reviewed in the marked-up version of the revised manuscript. I have also alerted my immediate colleagues to this matter and will inform the manuscript co-authors upon formal acceptance for publication. I will commit to using more appropriate vocabulary in future manuscripts, data presentations and communications.

Page 4 line 103 (original and revised unmarked manuscript): the typo “where” has been replaced with “were”.

Page 5 line 112 (original and revised unmarked manuscript): the selection process was not completely random and the criteria for selection have now been included. The priority was to recruit healthy subjects living with HIV, who were anti-retroviral naïve and willing to donate an extra blood sample for this study, which would not be completely random.

Page 9 line 195 (now line 199 of revised unmarked manuscript) and page 20 fig 2: The CMV and HIV peptides and other stimuli were not added together. To make this clearer the text has been modified to: “…. with one of the following stimuli;…” and fig 2 states “CMVpp65 or HIV-1 Nef”. The peptide concentration is re-stated on page 9 line 201 of the revised unmarked manuscript.

Page 20 line 395 (original manuscript): The term “spot forming units” and abbreviation (SFU) was stated on page 8 line 174 of the original and revised unmarked manuscript.

Fig 4: The typo testred has been corrected to tested.

Comment 6: I note the reviewer has elected to reveal their identity and that PLOS authors have the option to publish the peer review history. I agree to this option.

I have uploaded the original figure files to PACE. The original files met with PACE requirements but in reviewing the pdf previews I have decided to reduce the size of figures 2, 3 and 5 so as not to take up a full page for each figure. The revised figures have also been uploaded to PACE without messages indicating issues under “image problems” or “PACE adjustments”.

---

## [Editor Report · Decision Letter 1]

3 Nov 2021

Breadth of CD8 T-cell mediated inhibition of replication of diverse HIV-1 transmitted-founder isolates correlates with the breadth of recognition within a comprehensive HIV-1 Gag, Nef, Env and Pol potential T-cell epitope (PTE) peptide set

PONE-D-21-25896R1

Dear Dr. Hayes,

We’re pleased to inform you that your manuscript has been judged scientifically suitable for publication and will be formally accepted for publication once it meets all outstanding technical requirements.

Kind regards,

Douglas F. Nixon, M.D., Ph.D.

Academic Editor

PLOS ONE
---

## [Editor Report · Acceptance letter]

8 Nov 2021

PONE-D-21-25896R1 

Breadth of CD8 T-cell mediated inhibition of replication of diverse HIV-1 transmitted-founder isolates correlates with the breadth of recognition within a comprehensive HIV-1 Gag, Nef, Env and Pol potential T-cell epitope (PTE) peptide set 

Dear Dr. Hayes:

I'm pleased to inform you that your manuscript has been deemed suitable for publication in PLOS ONE. Congratulations! Your manuscript is now with our production department. 

Kind regards, 

on behalf of

Prof. Douglas F. Nixon 

Academic Editor

PLOS ONE